# An efficient blockchain-based authentication scheme with transferability

Xiushu Jin[1]*, Kazumasa Omote[2]

1 Graduate School of Science and Technology, University of Tsukuba, Tsukuba, Japan, 2 Faculty of Engineering, Information and Systems, University of Tsukuba, Tsukuba, Japan

* s2220550@u.tsukuba.ac.jp

## Abstract

In the development of web applications, the rapid advancement of Internet technologies has brought unprecedented opportunities and increased the demand for user authentication schemes. Before the emergence of blockchain technology, establishing trust between two unfamiliar entities relied on a trusted third party for identity verification. However, the failure or malicious behavior of such a trusted third party could undermine such authentication schemes (e.g., single points of failure, credential leaks). A secure authorization system is another requirement of user authentication schemes, as users must authorize other entities to act on their behalf in some situations. If the transfer of authentication permissions is not adequately restricted, security risks such as unauthorized transfer of permissions to entities may occur. Some research has proposed blockchain-based decentralized user authentication solutions to address these risks and enhance availability and auditability. However, as we know, most proposed schemes that allow users to transfer authentication permissions to other entities require significant gas consumption when deployed and triggered in smart contracts. To address this issue, we proposed an authentication scheme with transferability solely based on hash functions. By combining one-time passwords with Hashcash, the scheme can limit the number of times permissions can be transferred while ensuring security. Furthermore, due to its reliance solely on hash functions, our proposed authentication scheme has an absolute advantage regarding computational complexity and gas consumption in smart contracts. Additionally, we have deployed smart contracts on the Goerli test network and demonstrated the practicality and efficiency of this authentication scheme.

## 1 Introduction

Password authentication is one of the most common authentication schemes used in Internet applications, allowing access to various online services such as email, social media, online banking, and e-commerce websites. While password authentication is simple and convenient, it poses some security risks, such as password leaks, password guessing, and man-in-the-middle attacks. In comparison, public key authentication offers higher security. In public key authentication, users utilize a pair of keys: public and private keys. The user retains the public key, which can be publicly shared, while the private key is the user's personal information and

Blockchain-Based-Authentication-Scheme-with-Transferability.

**Funding:** Initials of the authors who received each award: X.J. Grant numbers awarded to each author: JP23K24844 and JP22K19768. The full name of each funder: Japan Society for the Promotion of Science URL of each funder website: https://www.jsps.go.jp/english/ Did the sponsors or funders play any role in the study design, data collection and analysis, decision to publish, or preparation of the manuscript?: No.

**Competing interests:** The authors have declared that no competing interests exist.

must be kept strictly confidential. When users attempt to log in, the system requires them to provide a digital signature generated using the private key, and only the corresponding public key can verify the signature successfully. With the rapid growth of internet technologies, there is an increasing demand for secure authentication schemes. Both password authentication and public key authentication face centralization issues, where the authentication scheme still relies on centralized authentication servers or databases to verify user identities. This centralized architecture carries the risk of a single point of failure, where an attack or failure of the authentication server could lead to service unavailability and user data leakage.

In practical applications, the authorization information must be invalidated after the authorized entity has utilized the authentication permission to prevent misuse, fraud, or unauthorized actions. This prevents the information from being reused or abused, which could lead to privacy breaches, identity theft, or other security risks for users. Therefore, promptly invalidating the authentication permission or ensuring that it is one-time-use only is crucial to provide security and user privacy. In previous research, the concept of one-time programs, initially proposed by Goldwasser et al. in 2008 [1, 2], addressed this issue by allowing programs to execute only once before "self-destructing." However, such schemes [3–6] rely on tamper-resistant hardware and secure memory devices, increasing hardware costs and complexity. The emergence of blockchain and smart contracts provides new solutions to this problem.

Given these considerations, blockchain technology is a primary solution to decentralize authentication schemes. Initially proposed by Satoshi Nakamoto in 2008, Blockchain [7] is a new distributed ledger technology that ensures tamper resistance, verifiability, and high availability. Smart contracts [8], initially proposed by cryptographer Nick Szabo in 1994, are computer protocols designed to execute, verify, or enforce contract terms in a digital environment. Ethereum [9] redefines smart contracts, enabling more flexible and powerful functionalities with applications in various domains such as cryptocurrency transactions, asset management, and decentralized applications (DApps). Ethereum smart contracts are executed by nodes on the Ethereum network, enhancing transparency, security, and reliability without relying on a centralized authority.

This paper presents a blockchain-based authentication scheme that innovatively addresses the security risks of authorization by combining one-time passwords and Hashcash. In this authentication scheme, smart contracts on the blockchain act as one-time key verifiers, representing multiple service providers in authenticating users. This undoubtedly reduces the burden on service providers to deploy authentication solutions and enables users to access various services with the same identity. Additionally, users can generate proof valid only for a specific agent and transfer the one-time key to that agent. Our authentication scheme ensures that only the designated agent can complete the authentication on the smart contract.

**Contributions**:

- Our authentication scheme allows users to authorize other agents while ensuring that only the designated agent can complete the user authentication process. Additionally, since the information representing the authorization is one-time-use only, it prevents the information from being reused or abused.

- Compared to other blockchain-based authentication schemes, our proposed scheme does not rely on zero-knowledge proofs or public-key algorithms; instead, it is solely based on combinations of hash functions, thus offering advantages in computational complexity and gas consumption in smart contracts.

The structure of this paper is outlined as follows: Section 2 provides the necessary background for the proposed scheme. Section 3 reviews related studies. Section 4 details the

proposed scheme. Section 5 presents the experimental setup and results. Section 6 conducts an analysis based on the findings from Section 5. Finally, Section 7 concludes the paper and suggests future research directions.

## 2 Prelimanaries

### 2.1 One-time password [10]

Lamport first proposed the concept of one-time passwords. Compared to static passwords, it is less susceptible to replay attacks. Their use can significantly increase security, as one-time passwords already used to log in to a service or execute a transaction are invalidated. New one-time passwords are dynamically generated and unpredictable. The user computes the secret's m times hash and sends the following $vk$ to the remote server.

$$vk \leftarrow H^m(p)$$

When the user requests login, a challenge $m - 1$ is given, and the user replies $v = H^{m-1}(p)$. The response received is $vk \overset{?}{=} H(v)$ is verified. If the verification is successful, update $vk$ as follows.

$$vk \leftarrow v$$

In our scheme, the one-time key ensures that the authorization is not enabled twice.

### 2.2 Hashcash [11]

Hashcash is the underlying technology of PoW and aims to curb the abuse of Internet resources. There are two principles of its operation:

1. Make mass-sending e-mails more expensive

2. Can only send e-mails slowly

When sending the e-mail, the sender must complete the $w$-bit partial hash collision calculation. This is because it consumes CPU operations and pays a cost. The cost function must satisfy three conditions. The conditions are that it is publicly auditable, consumes a certain number of resources, and is predictable.

A hash function $H$ with an output size of $k$ bits is publicly available. To send an e-mail, someone must obtain a random bit sequence $s$ and compute a nonce $x$ satisfying the following conditions.

$$\mathcal{T} \leftarrow \mathrm{MINT}(s, w) : \mathit{find}\ x \in_R \{0,1\}^*\ \mathrm{st}\ H(s\|x) \overset{\mathrm{left}}{=}_w 0^k$$

$$return(s, x)$$

Verification is relatively easy. With only one hash operation, the verifier can determine whether the nonce $x$ matches the requirement.

$$\mathcal{V} \leftarrow \mathrm{VALUE}(\mathcal{T}) : \quad H(s\|x) \overset{\mathrm{left}}{=}_v 0^k$$

$$return(v)$$

In this scheme, the Hashcash requires an attacker to calculate a nonce to validate the one-time key even if he steals it. As long as the actual agent's request reaches the smart contract before the attacker does, the attacker will not be granted authorization even if he calculates the corresponding nonce.

## 2.3 Blockchain and smart contracts

Bitcoin [7] uses mathematical calculations essentially similar to Hashcash and combines asymmetric cryptography, peer-to-peer transmission technology [12], and Hashcash algorithms to create a time-stamped data block called a blockchain, which is a distributed ledger. To enable more powerful programmable transaction logic, Ethereum updated the concept of smart contracts. A smart contract is a decentralized, automated computer program placed and executed on the blockchain. When certain trigger conditions are met, a set of mutually untrusted nodes automatically execute a predefined smart contract without needing a trusted third party.

Smart contracts have the general characteristics of a blockchain, which employs a decentralized storage scheme. Furthermore, the results of the execution of smart contracts are highly trustworthy, as all blockchain network nodes verify them. Therefore, trust-secured transactions can be processed without a trusted third party.

# 3 Related work

## 3.1 Blockchain-based authentication and ZKP

Bai et al. [13] proposed the Health-zkIDM system, which addresses the user authentication challenges in online medical services. Their system leverages decentralized user authentication based on zero-knowledge proof (ZKP) [14] and blockchain technology. The centralized identity management systems used in healthcare often limit interoperability between institutions and raise privacy concerns. Feng et al. [15] proposed a theoretical solution for Self-Sovereign Identities (SSI) using zero-knowledge proof protocols. Their approach addresses the need for secure and decentralized identity management. By employing ZKP protocols based on the discrete logarithm difficulty and introducing automorphism group properties, the proposed SSI protocol enables the creation and management of identities with minimal disclosure of information to trusted parties. This protocol ensures compliance with regulations such as eIDAS and GDPR while enhancing privacy and security.

Dieye et al. [16] conducted a comprehensive survey on integrating zero-knowledge proof technology into blockchain for secure identity-sharing systems. Their paper critically evaluates existing literature and identifies challenges and opportunities in this domain. By examining the assimilation of ZKP technology into the blockchain, Dieye et al. highlight the potential for enhanced privacy and security in identity-sharing systems. Diro et al. [17] summarized the contributions of the aforementioned authors, along with others in related fields. They highlighted various applications of zero-knowledge proof technology in enhancing security and privacy in different contexts, such as smart city resources, Internet of Things (IoT) systems, authentication architectures, and continuous authentication solutions. These applications aim to address the vulnerabilities inherent in identity-sharing systems and improve data security and privacy in online services.

## 3.2 Blockchain-based token authentication

The single-use proxy signature proposed by Krenn et al. [18] has a software-based feature. This feature avoids dedicated hardware support compared to schemes that require tamper-proof hardware and a trusted execution environment [19–21]. The scheme is based on the blockchain and verifies the transfer utilizing zero-knowledge proofs in the communication. The blockchain's tamper-resistance can be used to ensure that the threshold for the number of transfers set by the user is not exceeded. The agent inspires the smart contract and verifies the transfer.

Zhang et al. [22] propose a blockchain-based authentication scheme with a one-time password. It was suggested to solve the problem of establishing trust between two unknown entities in traditional authentication, which requires a trusted third party. Specifically, the scheme uses Park's one-time password [23] scheme to record the next commitment value for each one-time password. The user signs the one-time password so the agent cannot forge the signature, and the commitment value cannot be tampered with. As only the user knows the one-time password corresponding to the commitment value and the agent does not have the user's private key, the transferred permission is limited to the number of times.

In other blockchain authentication schemes, Alharbi et al. [24] propose a two-factor authentication scheme, adding a security layer at the user's login. The authentication process is completed using a one-time password generated by a smart contract, which is also sent to the application or website using its hash value. Catalfamo et al. [25] propose a distributed micro-services and blockchain-based one-time password (MBB-OTP) scheme, which uses three loosely coupled and lightweight services to mitigate the threat of DoS. The results of the comparison are shown in Table 1. The criteria for comparison are as follows.

- **C1**. Is the authorization system contemplated?

- **C2**. Is the authorization system possible?

- **C3**. Is the study based only on hash functions?

- **C4**. Is it possible not to transfer the master private key?

- **C5**. Is it possible to limit the number of times the permission can be transferred?

- **C6**. Is it possible to resist replay attacks?

According to **C1** and **C2**, **KL21** and the proposed scheme take into account the authorization system. In contrast, **ZWY19**, **AA19**, and **CRC21** do not consider transferring authentication permissions. However, they can still achieve authorization by directly transferring the information for authentication permission and can serve as control groups in comparing computational complexity and gas consumption. From **C3**, the proposed scheme only utilizes hash functions. Therefore, the proposed scheme outperforms the existing studies regarding computational complexity and gas consumption. From **C4** and **C5**, **KL21**, **ZWY19**, the proposed scheme can transfer the authentication permission without transferring the user's master key and ensure that the authentication permission will not be reused. In contrast, **AA19** and **CRC21** require transferring the master key, as the user does not directly generate the one-time password. From **C6**, apart from the proposed scheme, existing studies tend to use public key algorithms to prevent replay attacks. In contrast, the proposed method employs an approach similar to Hashcash to resist replay attacks.

**Table 1. Comparison of blockchain-based authentication schemes.**

|  | C1 | C2 | C3 | C4 | C5 | C6 |
|---|---|---|---|---|---|---|
| **KL21** [18] | ✓ | ✓ |  | ✓ | ✓ | ✓ |
| **ZWY19** [22] |  | ✓ |  | ✓ | ✓ | ✓ |
| **AA19** [24] |  | ✓ |  |  |  | ✓ |
| **CRC21** [25] |  | ✓ |  |  |  | ✓ |
| Proposed scheme | ✓ | ✓ | ✓ | ✓ | ✓ | ✓ |

### 3.3 Blockchain-based authentication and IoT

Asif et al. [26] introduced a Blockchain-based security scheme tailored for secure authorized access to smart city resources. This scheme combines the ACE framework-based authorization Blockchain and the OSCAR object security model to enable flexible and trustless authorization schemes while utilizing a public ledger for structuring multicast groups. A user-friendly meteor-based application was also developed for heterogeneous smart city technologies, allowing users to interact with and control various resources such as traffic lights and surveillance cameras. Al Hwaitat et al. [27] proposed a unique approach for a massive IoT system based on a permissions-based blockchain, addressing complexity and storage overhead issues. Their scheme integrates homomorphic encryption for IoT data encryption and cloud upload, offering scalability and optimized storage. They demonstrated superior performance to benchmark frameworks through extensive simulation results, highlighting enhanced security and privacy in IoT services.

Khashan et al. [28] presented a hybrid centralized and blockchain-based authentication architecture for IoT systems. Their approach leverages edge servers for centralized authentication while establishing a blockchain network of centralized edge servers for decentralized authentication across heterogeneous IoT systems. Lightweight cryptographic schemes ensure efficient authentication, addressing resource constraints inherent in IoT devices. Al-Naji et al. [29] proposed a distributed and scalable continuous authentication scheme implemented using blockchain technology. This solution incorporates fog nodes to address IoT resource limitations, introducing a trust module utilizing face recognition machine learning to detect abnormal access.

Kudva et al. [30] discussed the revolutionary impact of ride-hailing services on the transportation industry worldwide but pointed out that these services are highly centralized, controlled by central authorities, and maintain ownership of user data. Such centralized platforms raise concerns about service policies and data reliability, as all data records would be lost or compromised in the event of central server data tampering or ransomware attacks. The study explores the possibility of implementing decentralized applications in ride-hailing services. It proposes a blockchain-based ride-hailing service architecture with user authentication and ride-scheduling functionalities. Shawky et al. [31] proposed a scheme for group key distribution using smart contract-based blockchain technology. The functions of the smart contract enable secure distribution of the group session key following initial legitimacy detection using public key infrastructure-based authentication. They propose a lightweight symmetric key cryptography-based group signature method for message authentication, supporting verification of vehicular ad hoc networks (VANETs).

## 4 Constructions

### 4.1 Overview

This section introduces the scheme of transferring authentication permission. The specific flow is shown in Fig 1. (1) The user stores the verification key $vk$ in the blockchain through the smart contract. (2)The user securely sends the agent the one-time key corresponding to $vk$. (3) To obtain authorization, the agent provides a one-time key and proof of identity as the agent to the smart contract. (4) The smart contract finds the corresponding $vk$ in the blockchain, and (5) automatically verifies the above one-time key and proof. (6) If the Verification succeeds, it outputs OK; if it fails, it invalidates the current one-time key. Due to the transparency of the blockchain network, users, proxies, and trusted third parties can check at any time whether authorization and authentication have been successful.

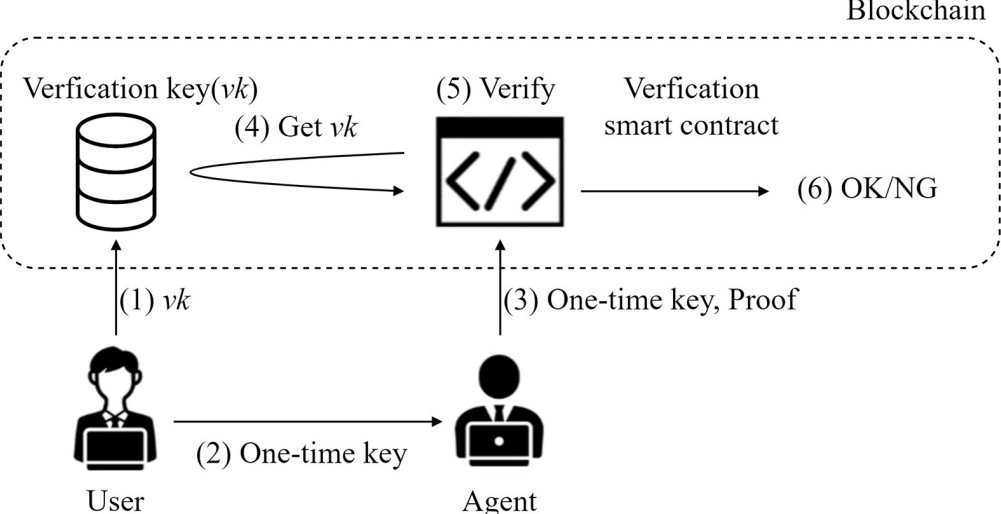

**Fig 1. Overall diagram of the proposed scheme.**

## 4.2 Entities

**User**: Provider of the transferred authentication permission. The user stores the verification key in the blockchain using a smart contract, generates the corresponding one-time key, and sends the one-time key to the agent to transfer the permission.

**Agent**: The recipient of the permission. Receives the user's one-time key, calculates the proof associated with its address, and sends the one-time key and the proof to the Verification smart contract.

**Verification Smart Contract**: A process for automatically verifying the authentication of an agent. The Verification Smart Contract verifies the agent's legitimacy by utilizing the proof and the one-time key.

## 4.3 Notation

Table 2 shows the notation used in this paper.

## 4.4 Attack model

**Assumption**

- Assume that the user and the agent are trustworthy, and an external attacker is assumed as the attacker.

**Table 2. Notation.**

| Notification | Description |
| --- | --- |
| $p$ | User's master key |
| $H$ | Cryptographic hash function |
| $w$ | Number of bits in partial hash collisions |
| $m$ | Maximum number of authentication |
| $n$ | The $n$th authentication. |
| $vk$ | Verification key in smart contracts |
| $addr$ | Wallet address of the agent |

- Assume that no master or one-time keys are stolen from the user's or agent's terminal.

- Assume that the communications between users and agents have appropriate security measures, such as using protocols like TLS.

**Threat**

Based on the assumptions above, the proposed scheme assumes that the user and the agent do not engage in malicious operations and that a secure communication channel exists between them. Additionally, due to the decentralized structure of the blockchain, tampering with data on the blockchain or programs on smart contracts is nearly impossible. Therefore, the primary risks faced by the system proposed in this paper are eavesdropping and impersonation, targeting communication between the agent and the smart contract.

During the stage illustrated as (3) in Fig 1, attackers could potentially eavesdrop on the communication between the agent and the smart contract while the agent sends the one-time key and proof. This interception could grant them the ability to impersonate legitimate identities. Specifically, attackers might intercept the transaction initiated by the agent, thus gaining access to the one-time key. Subsequently, to fabricate the proof, attackers would utilize their address along with the pilfered one-time key to generate a nonce. Ultimately, attackers would dispatch a transaction to the smart contract and carry out a replay attack to gain authorization from the smart contract.

## 4.5 Scheme

The complete structure and procedure of the authentication permission transfer scheme are shown in Fig 2. It is assumed that the verification smart contract has already been deployed on the blockchain.

1. **Send master verification key**: the user calculates $\{vk = H^{m+1}(p)\}$ and stores it in the blockchain via smart contract using a transaction.

2. **Transfer of permission (one time)**: the user computes $(m + 1 - n)$ times hash $\{vk' = H^{m+1-n}(p)\}$ for $p$, i.e. the $n$th one-time key, and sends it securely to the agent. In this case, $\{vk = H^{m+2-n}(p)\}$.

3. **Authentication request**: the Hashcash is used here. Specifically, the agent seeks a value of nonce such that the hash for the concatenation of his wallet address and nonce $H(addr\|nonce)$ and $H^{m+1-n}(p)$ collide by $w$ bits on the left side as in the following formula.

$$nonce \in_R \{0, 1\}^*$$
$$\text{st } H(addr\|nonce) \stackrel{\text{left}}{=}_w H^{m+1+n}(p)$$

The agent then provides the smart contract with the verification message $\{vk' = H^{m+1-n}(p), nonce\}$. This *nonce* is the proof.

4. **Execution of the verification smart contract**: When the smart contract receives the nonce and the one-time key, it verifies that the one-time key and the hash for the concatenation of the address and the nonce have a left $w$-bit collision:

$$H(addr\|nonce) \stackrel{\text{left?}}{=}_w vk'$$

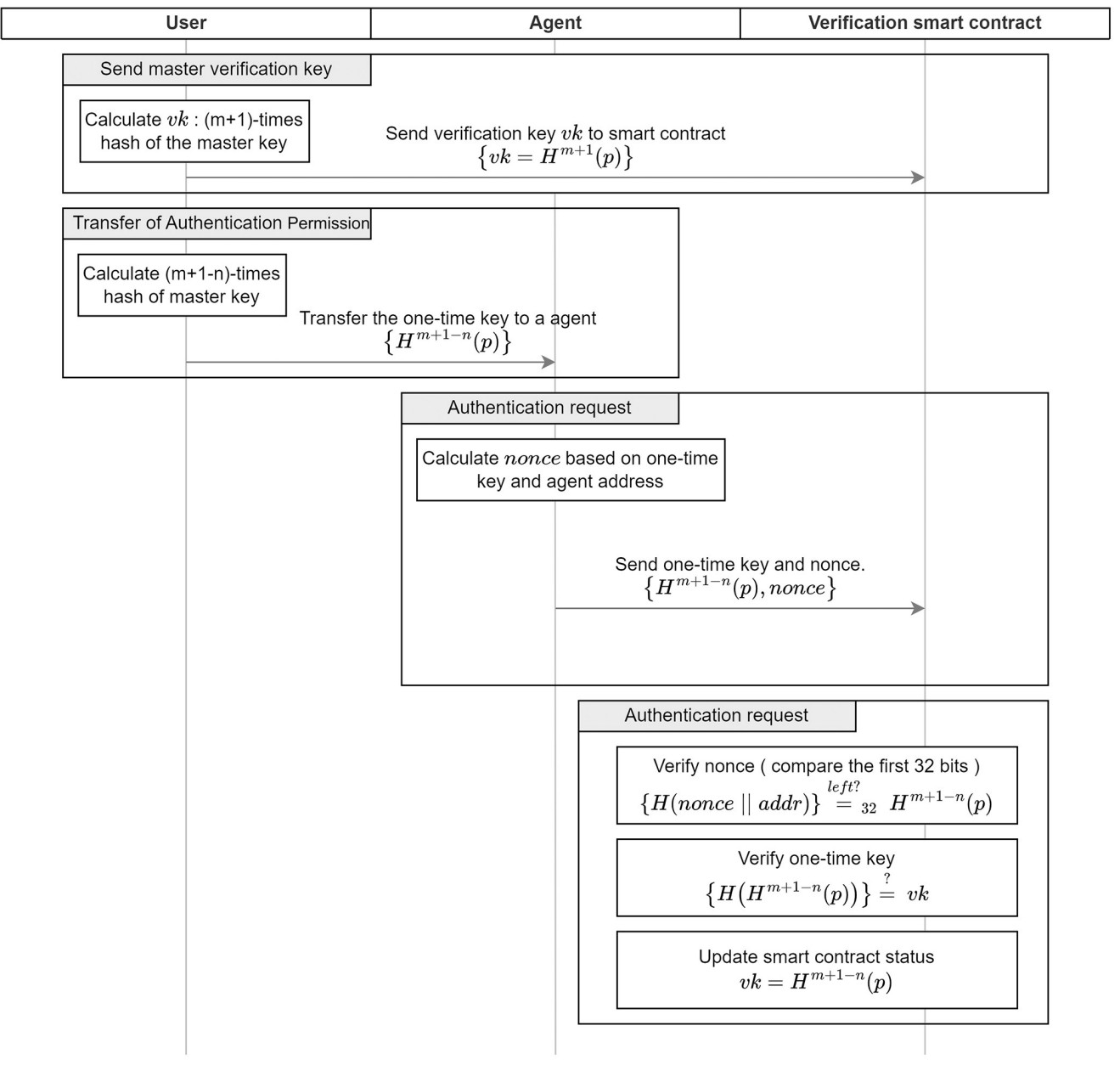

**Fig 2. Flow diagram of the proposed scheme.**

The one-time key is verified as follows:

$$H(vk') \stackrel{?}{=} vk$$

The verification smart contract then updates $vk$ from $H^{m+2-n}(p)$ to $H^{m+1-n}(p)$.

## 5 Evaluation

### 5.1 Purpose

This evaluation aims to identify the authentication costs and demonstrate its feasibility. Specifically, we implement a verification smart contract and measure the gas consumption of its

deployment, the processing time and gas consumption of verification, the gas consumption of the verification key registration, and the time required to create the proof.

## 5.2 Experimental environment

The verification smart contract is implemented in the Solidity language and deployed to Goerli, the Ethereum blockchain testnet, via the Remix kit. The hash algorithm used is SHA256. The verification processing time was measured on a local environment (Ryzen 5 4600H 3.00 GHz CPU, 16.0 G RAM).

## 5.3 Verification smart contract

Algorithm 1 provides the algorithm for verifying smart contracts. Specifically, the proxies provide their wallet address when sending the transaction, which is used to verify the legitimacy of the nonce. If the nonce verification fails, FALSE is returned. If the nonce verification succeeds and the hash value of the one-time key is equal to $vk$, $vk$ is updated, and TRUE is returned.

**Algorithm 1: Verification smart contract**

```
Input: nonce, addr, vk'
Output: true or false

if H(nonce, addr) ≠³²left vk' then
  return false;
else
  if H(vk') = vk then
    vk = vk';
    return true;
  else
    return false;
  end
end
```

## 5.4 Deployment and verification costs

The cost of a smart contract depends on the computation required. To efficiently use smart contracts, it is necessary to consider gas consumption. Therefore, this section calculates the gas needed for deploying the scheme, the calculation time, and the verification process. Table 3 shows the gas consumption a user requires to deploy a smart contract, the calculation time, and the gas consumption required for an agent to complete the transfer. We used the Ethereum Virtual Machine and the Ethereum testnet Goerli to calculate the theoretical gas needed for deploying and verifying the smart contract. The results show that deploying the smart

**Table 3. Deployment and verification costs of blockchain-based authentication schemes.**

|  | Deployment | Verification | |
|---|---|---|---|
|  | Gas consumption | Time | Gas consumption |
| Proposed scheme | 304.6kgas | 0.0039ms | 29.9kgas |
| **ECDSA** [32] | 333.5kgas | 2.7192ms | 35.9kgas |
| **SJ23** [31] | 263.5kgas | - | 69.0kgas |
| **KL21** [18] | 534.0kgas | - | 108.0kgas |
| **KN20** [30] | 1240.3kgas | - | 244.5kgas |
| **ZWY19** [22] | - | 0.548 ms | - |

contract consumes 304.5 kgas, and verification consumes 29.8 kgas. Additionally, the time required to generate a hash chain in the local environment is 1.5371 ms.

In this evaluation, we included a baseline comparison with an authentication scheme using ECDSA in addition to existing research. Table 3 shows that the user's Deployment gas consumption in our scheme is 304.6 kgas, 28.9 kgas lower than ECDSA. Regarding the time taken for verification, the time required to complete authentication under the scheme is 0.0039 ms, 697 times faster than ECDSA. The gas consumption of verification is required for the agent to completethe transaction process once. In our scheme, the gas consumption is 29.9 kgas, 6.0 kgas lower than ECDSA. Since **ZWY19** has not provided gas consumption data on the Ethereum testnet, we only compare the time required to complete verification. Compared to other authentication schemes on the blockchain, our scheme also requires less gas to execute the verification operations, far less than similar methods such as **SJ23**, **KL21**, and **KN20**. Based on this, our scheme has certain advantages in terms of time complexity or gas consumption compared to existing authentication schemes on the blockchain.

## 5.5 Time taken to create the proof

The proof is necessary to secure the transfer of authentication permission. A replay attack could be successful if an attacker sniffs the one-time key from the transaction, generates a new proof, and sends it to the smart contract. Therefore, the time to create proof must be sufficiently long to prevent replay attacks. This section provides the time required to generate the nonce used as proof in our scheme.

As shown in Fig 3 and S1 File.Dataset, the nonce creation was performed 100 times with $w$ = 32. As shown in Table 4, The longest time is 20373.5s; the shortest time is 1677.83s, and the average time is 9151.92s. From this, without considering network latency, on average, an attacker would require approximately two hours to forge a nonce and execute a replay attack. In contrast, a legitimate agent can complete one authentication in an average of only 0.0039 ms, significantly less than two hours. When the attacker's authentication request arrives, the smart contract that has already processed the agent's request will reject the attacker's request.

## 6 Discussion

### 6.1 Replay attack

Suppose there exists a commitment value on the smart contract $\{vk = H^{m+1}(p)\}$. When user Alice performs the $n$th authorization, Alice sends to the agent Bob a hash value obtained by hashing the master key p for m+1-n times, i.e., $\{vk' = H^{m+1-n}(p)\}$. Based on the assumptions made in our attack model in Section 4.4, the attacker cannot directly steal the master key p from user Alice's terminal, nor can they eavesdrop on the secure channel between user Alice and the agent Bob, meaning they cannot intercept $vk'$ before Bob initiates the authentication request to the smart contract. Therefore, we consider the only scenario for replay attacks to exist as outlined in Step 3 of Section 4.5, where the agent Bob completes the nonce calculation and sends the verification message $\{vk' = H^{m+1-n}(p), nonce\}$ to the smart contract. At this point, upon intercepting the verification message $\{vk' = H^{m+1-n}(p), nonce\}$, the replay attack can be divided into two cases.

In the first case, if the attacker obtains the verification message that has already been verified by the smart contract, i.e., the attacker obtains $\{vk' = H^{m+1-n}(p), nonce\}$, while the commitment value stored in the smart contract at that time is $\{vk = H^{m+1-n}(p)\}$, due to the preimage resistance of the hash function, the attacker cannot derive $H^{m-n}(p)$ from $H^{m+1-n}(p)$, and they also lack the corresponding nonce. Therefore, in this case, the replay attack cannot be completed.

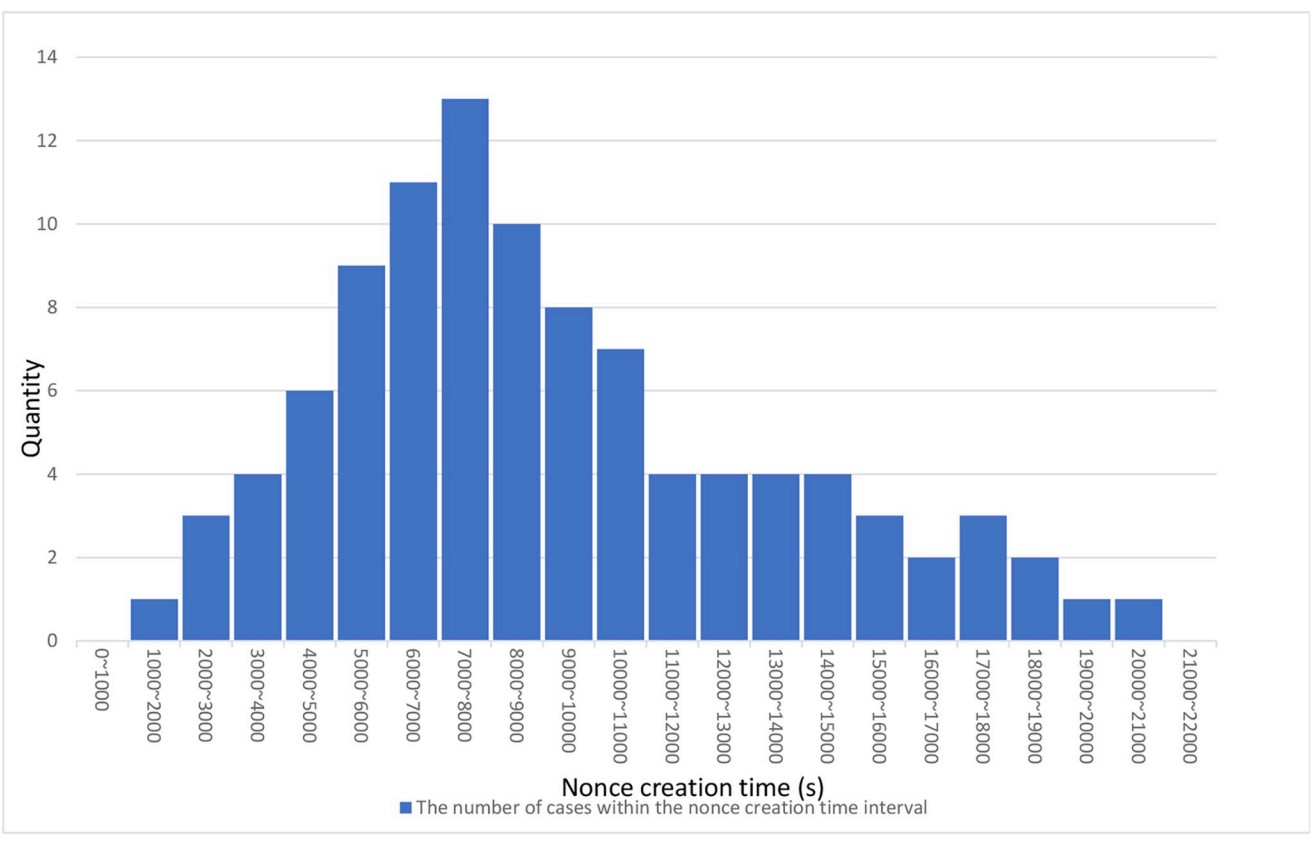

**Fig 3. Distribution of the time taken to create a proof.**

In the other case, the attacker manages to intercept the verification message sent by the agent Bob. If the attacker directly sends the verification message $\{vk' = H^{m+1-n}(p), nonce\}$ to the smart contract at this time, it will be rejected by the smart contract. As shown in Step 3 of Section 4.5, the nonce in the message must match the attacker's address for the smart contract to recognize it. In Section 5.5, our test results demonstrate that such attempts require an average of at least an hour of trying, while a single transaction typically completes confirmation within seconds to minutes. Agent Bob has sufficient time to send the verification message through an unintercepted channel and warn user Alice and other entities of unauthorized operations to mitigate subsequent risks. Therefore, the replay attack still cannot succeed.

## 6.2 Forgery of proof

According to Section 6.1, assuming the attacker has obtained the authentication message sent by the agent to the smart contract, i.e., the attacker has acquired $\{H^{m+1-n}(p), nonce\}$. However, since the nonce generated by the agent serves as proof of the agent's wallet address, the attacker needs to create a new *nonce* belonging to their wallet address. We simulated this scenario, as

**Table 4. Nonce creation time.**

| Average | Max | Min |
|---|---|---|
| 9151.92s | 20373.5s | 1677.83s |

shown in Section 5.5. Notably, the Poisson distribution serves as a close approximation to the time needed for nonce generation. In our testing, the shortest time required to create proof was 1677.83 seconds. Based on our results, the probability of directly forging a new proof nonce within seconds to minutes, the typical time frame for completing a transaction, is extremely low. Suppose the smart contract accepts the authentication request sent by the agent before this proof creation time. In that case, the verification smart contract will reject subsequent authentication requests sent by the attacker. Furthermore, even if the verification message sent by the agent is intercepted and cannot communicate with the smart contract, based on the assumptions made in Section 4.4 of the attack model, the agent can still notify the user of possible unauthorized authorizations in advance, enabling them to prepare and mitigate potential risks.

### 6.3 Efficiency

In real applications, the number of users requiring authorization will likely be high. Having each user deploy a new smart contract incurs enormous registration gas consumption. In this case, recording multiple *vk* in one smart contract is sufficient. The registration gas consumption paid by individual users can be reduced. However, the gas consumption for verification generally increases as the number of verification keys increases. We implemented key retrieval in the smart contract using mapping. In this scenario, while increasing the number of users does not incur additional computational overhead, the cost of a single verification has increased from 29.9 kgas to 49.8 kgas.

### 6.4 vk update v.s. vk fix

Depending on the number of user transfers of authentication permission, the following two *vk* policies are possible. Our scheme employs *vk* update.

- *vk* update: Update the *vk* recorded in the smart contract each time verification is performed.

- *vk* fix: Do not update the *vk* recorded in the smart contract.

The *vk* fix does not update *vk*, saving gas consumption for blockchain operations. However, as *n* increases, more hash operations are required per verification.

$$H^n(H^{m+1-n}(p)) \stackrel{?}{=} vk \tag{6}$$

However, it has the advantage of reducing the initial gas consumption, as it reduces the number of modification operations on the variables stored in the smart contract.

On the other hand, the proposed scheme with the policy of *vk* update requires only one hash operation per verification. Therefore, even if *n* increases, the hash operation is constant, which can avoid increased gas consumption.

Fig 4 displays the gas consumption required for ten verifications under two policies. Using the *vk* fix, the gas consumption is lower than the *vk* update for the first two verification operations but higher than the *vk* update for three or more. Therefore, the *vk* update is desirable in many cases where the number of user delegations is more than three. Hence, we have adopted the *vk* update.

## 7 Conclusion

This paper proposes an efficient authentication scheme applicable to blockchain systems. The scheme relies solely on hash functions and can limit the usage of authorizations. We have demonstrated that the proposed scheme is available, decentralized, and can resist attacks. Since the

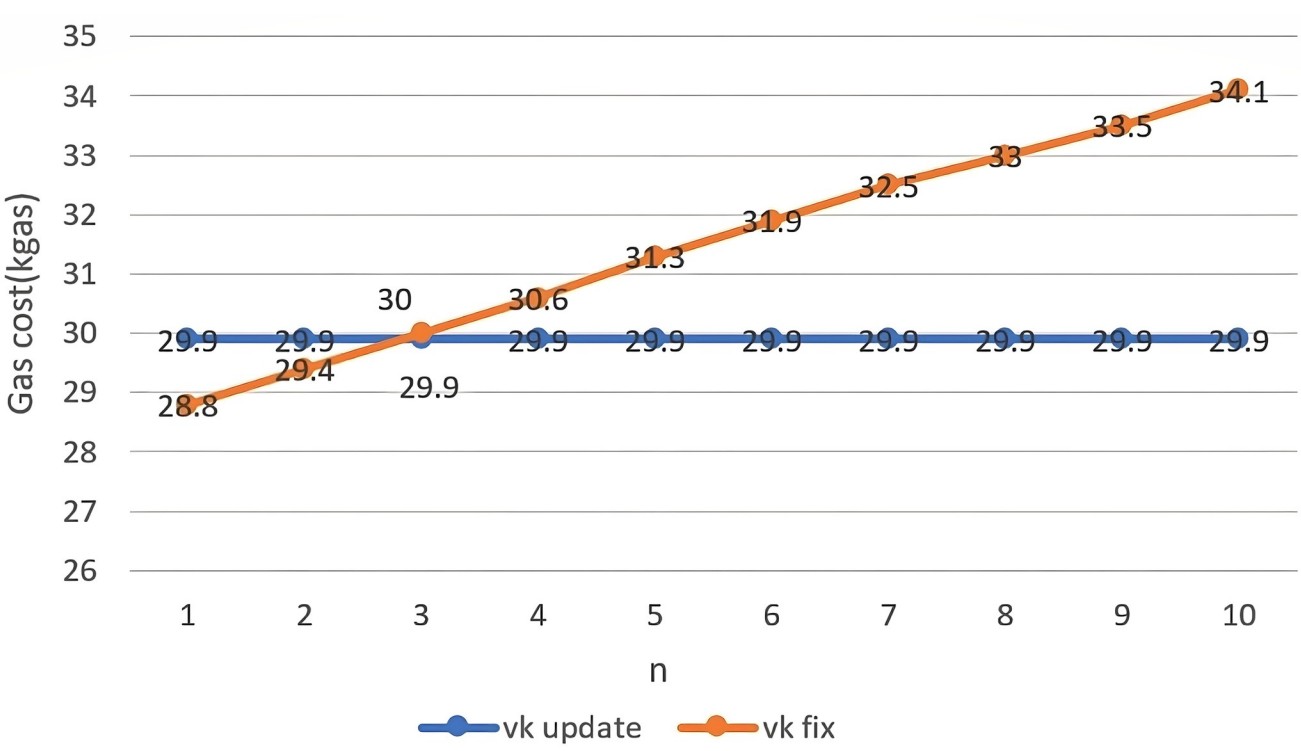

**Fig 4. Costs of basic and extended schemes.**

authentication process solely relies on hash functions, it can efficiently execute the verification process of smart contracts, thereby reducing computational complexity and gas consumption. In our evaluation, our scheme consumes less than one-third of the gas compared to existing schemes. The smart contract implementation presented in this study was conducted on the Ethereum test network Goerli, showcasing the feasibility of the proposed scheme. In future work, we will optimize the blockchain's network parameters and underlying structure to achieve a more efficient authentication scheme.

## Supporting information

**S1 File. Dataset.**
(XLSX)

## Author Contributions

**Conceptualization:** Xiushu Jin, Kazumasa Omote.

**Data curation:** Xiushu Jin.

**Formal analysis:** Xiushu Jin.

**Funding acquisition:** Kazumasa Omote.

**Investigation:** Xiushu Jin, Kazumasa Omote.

**Methodology:** Xiushu Jin, Kazumasa Omote.

**Project administration:** Kazumasa Omote.

**Resources:** Xiushu Jin, Kazumasa Omote.

**Software:** Xiushu Jin.

**Supervision:** Kazumasa Omote.

**Validation:** Xiushu Jin.

**Visualization:** Xiushu Jin.

**Writing – original draft:** Xiushu Jin.

**Writing – review & editing:** Xiushu Jin, Kazumasa Omote.

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
