## [Decision Letter · Decision Letter 0]

22 May 2024

PONE-D-24-15013An Efficient Blockchain-Based Authentication Scheme with TransferabilityPLOS ONE

Dear Dr. JIN,

Thank you for submitting your manuscript to PLOS ONE. After careful consideration, we feel that it has merit but does not fully meet PLOS ONE’s publication criteria as it currently stands. Therefore, we invite you to submit a revised version of the manuscript that addresses the points raised during the review process.

We look forward to receiving your revised manuscript.

Kind regards,

Hasan Tahir

Academic Editor

PLOS ONE

Journal Requirements:

2. Please note that PLOS ONE has specific guidelines on code sharing for submissions in which author-generated code underpins the findings in the manuscript. In these cases, all author-generated code must be made available without restrictions upon publication of the work. 

Please review our guidelines at https://journals.plos.org/plosone/s/materials-and-software-sharing#loc-sharing-code and ensure that your code is shared in a way that follows best practice and facilitates reproducibility and reuse.

Securing Fog Computing with a Decentralised User Authentication Approach Based on Blockchain - https://doi.org/10.3390/s22103956

(among others)

In your revision ensure you cite all your sources (including your own works), and quote or rephrase any duplicated text outside the methods section. Further consideration is dependent on these concerns being addressed.

4. We note that your Data Availability Statement is currently as follows: 

"All relevant data are within the manuscript and its Supporting Information files."

5. Please include a copy of Table 6 which you refer to in your text on page 11.

**Additional Editor Comments:**

Please incorporate the changes suggested by our worthy reviewers. Overall the paper flow and clarity can be improved. Language of the paper can also be improved in many places.

Reviewers' comments:

Reviewer's Responses to Questions

**Comments to the Author**

1. Is the manuscript technically sound, and do the data support the conclusions?

Reviewer #1: Partly

Reviewer #2: Yes

2. Has the statistical analysis been performed appropriately and rigorously? 

Reviewer #1: No

Reviewer #2: Yes

3. Have the authors made all data underlying the findings in their manuscript fully available?

Reviewer #1: No

Reviewer #2: Yes

4. Is the manuscript presented in an intelligible fashion and written in standard English?

Reviewer #1: No

Reviewer #2: Yes

5. Review Comments to the Author

Reviewer #1: • The manuscript's structure was not well-prepared.

• The contribution in this manuscript is simple and has little novelty.

• Although the authors focused on the gas computation factor in their proposed system, they have not compared their system with those mentioned in the Related Works section.

• Some figures in the manuscript are not clear and the titles of their axis need to be clarified in the figures themselves.

• Some sections in the manuscripts don't contain any paragraphs such as (4.3 Notation)

• The study relied on three assumptions, some of which, like the third assumption, cannot be achieved in the real world.

• The results obtained should be presented in a more comprehensible and lucid manner.

Reviewer #2: Hope its a preliminary research work, which has been well articulated.

Authors are asked to incorporate or explain the same with any one of the suitable application. Also list few suitable applications where this scheme can be incorporated.

This work mainly focus on hashing concept and computational complexity how this scheme will be helpful or suitable while scaling up the experiment in terms of no. of users ? Justify.

6. PLOS authors have the option to publish the peer review history of their article (what does this mean?). If published, this will include your full peer review and any attached files.

Reviewer #1: No

Reviewer #2: No

---

## [Author Response · Author response to Decision Letter 0]

25 Jul 2024

Dear Reviewers and Editor,

Thank you for your valuable feedback on my manuscript. I have carefully considered your comments and have addressed them in detail in the attached Word document "Response to Reviewers." Please find my responses and revisions outlined therein.

Please do not hesitate to reach out if you have any further questions or require clarification on any points. Your input has been greatly appreciated, and I look forward to your continued guidance through the review process.

Best regards,

Xiushu Jin

---

## [Decision Letter · Decision Letter 1]

26 Aug 2024

An Efficient Blockchain-Based Authentication Scheme with Transferability

PONE-D-24-15013R1

Dear Dr. JIN,

We’re pleased to inform you that your manuscript has been judged scientifically suitable for publication and will be formally accepted for publication once it meets all outstanding technical requirements.

Kind regards,

Hasan Tahir

Academic Editor

PLOS ONE

Additional Editor Comments (optional):

Paper is recommended for acceptance. While authors have improved the paper I will ask them to please review/ examine the paper once more before it goes into final publishing.

Reviewers' comments:

Reviewer's Responses to Questions

**Comments to the Author**

1. If the authors have adequately addressed your comments raised in a previous round of review and you feel that this manuscript is now acceptable for publication, you may indicate that here to bypass the “Comments to the Author” section, enter your conflict of interest statement in the “Confidential to Editor” section, and submit your "Accept" recommendation.

Reviewer #2: All comments have been addressed

2. Is the manuscript technically sound, and do the data support the conclusions?

Reviewer #2: Yes

3. Has the statistical analysis been performed appropriately and rigorously? 

Reviewer #2: Yes

4. Have the authors made all data underlying the findings in their manuscript fully available?

Reviewer #2: Yes

5. Is the manuscript presented in an intelligible fashion and written in standard English?

Reviewer #2: Yes

6. Review Comments to the Author

Reviewer #2: (No Response)

7. PLOS authors have the option to publish the peer review history of their article (what does this mean?). If published, this will include your full peer review and any attached files.

Reviewer #2: No

---

## [Editor Report · Acceptance letter]

30 Aug 2024

PONE-D-24-15013R1 

PLOS ONE

Dear Dr. JIN, 

I'm pleased to inform you that your manuscript has been deemed suitable for publication in PLOS ONE. Congratulations! Your manuscript is now being handed over to our production team.

Kind regards, 

on behalf of

Dr. Hasan Tahir 

Academic Editor

PLOS ONE